# Mortality and comorbidities among teaching professionals: A cross-sectional study in Colombia

**Daniela Sánchez-Santiesteban**[1,2]*, **Hernando Bayona-Rodríguez**[3], **Giancarlo Buitrago**[1,2]

**1** Instituto de Investigaciones Clínicas, Facultad de Medicina, Universidad Nacional de Colombia, Bogotá, Colombia, **2** Fundación Cardioinfantil - Instituto de Cardiología, Bogotá, Colombia, **3** Facultad de Ciencias Económicas, Universidad Nacional de Colombia, Bogotá, Colombia

* dansanchezsa@unal.edu.co

## Abstract

### Background

Teachers play a critical role in social and economic development, yet evidence on their health outcomes in Latin America remains scarce. In Colombia, teachers are generally classified in occupational risk level 1, a category considered to have minimal hazards. This study aimed to describe and compare mortality and comorbidities among teachers and non-teachers in the same risk category, and to explore differences across educational levels within the teaching profession.

### Methods

We conducted a retrospective cohort study using four linked national administrative databases in 2017. Adults affiliated to the contributory health insurance scheme and classified under occupational risk level 1 were included. Teachers were identified and stratified by educational level. Outcomes included one-year all-cause mortality and prevalence of mental health and hearing disorders. Multivariable logistic regression models adjusted for sociodemographic and clinical covariates were used to estimate associations.

### Results

A total of 4,256,719 individuals were included, of whom 353,985 (8.3%) were teachers. Teachers were older (mean age 40.1 vs 36.4 years) and more frequently female (69% vs 60%) than non-teachers. The one-year mortality proportion was higher among teachers (0.14%) than non-teachers (0.11%). After adjustment, teaching was associated with a 15% higher risk of mortality (OR: 1.15, 95% CI: 1.03–1.28). No significant associations were found for mental health (OR: 0.98, 95% CI: 0.96–1.01) or hearing disorders (OR: 0.97, 95% CI: 0.92–1.02). Subgroup analyses showed

**Data availability statement:** The study uses four anonymized administrative databases provided by the Office of Information Technologies of the Colombian Ministry of Health and Social Protection for research purposes. These databases include the Wages Database (PILA), the Capitation Unit Sufficiency Study Database (UPC), the Unique Enrollees Database (BDUA), and the Mortality Registry Module (RUAF). Due to institutional restrictions, these data are not publicly available. Researchers may request access directly from the Ministry's Technology of Information and Communication Office at correo@minsalud.gov.co.

**Funding:** The author(s) received no specific funding for this work.

**Competing interests:** The authors have declared that no competing interests exist.

the highest mortality proportions among teachers in technical and technological education.

## Conclusions

Despite being classified in the lowest occupational risk level, private-sector teachers in Colombia exhibited higher mortality compared with other workers in the same group. Differences in mental health and hearing disorders were not significant. These findings highlight the need to strengthen surveillance, prevention, and protection strategies tailored to teachers, recognizing them as a priority population within occupational health and education policies.

## Introduction

Teaching is widely recognized as a cornerstone profession for social and economic development [1]. Beyond the transmission of knowledge, it shapes the foundations of human capital, influences workforce productivity, and contributes to reducing social inequalities [2,3]. Teachers are essential agents whose performance and well-being directly affect students' learning outcomes and the overall effectiveness of educational systems, which in turn have long-term implications for economic growth and social mobility [4–7].

Globally, research has shown that teachers may present distinct morbidity profiles relative to other workers, with a higher prevalence of mental health problems and hearing disorders, often linked to occupational stress and prolonged exposure to noisy environments [8–11]. Teachers are also exposed to physical and environmental hazards that contribute to other morbidities, such as ergonomic strain, poor air quality, and inadequate acoustic conditions further aggravate physical health outcomes [12]. In Latin America, growing evidence underscores a substantial burden of occupational morbidity among teachers. Studies from Brazil, Mexico, and Chile have consistently reported high rates of burnout, anxiety, and voice and hearing disorders associated with overcrowded classrooms, noise, and precarious working environments [13–15]. In Colombia, similar patterns have been observed, identifying a high prevalence of burnout syndrome, anxiety, and musculoskeletal complaints in the education sector [15–19]. Mortality patterns among teachers have been relatively understudied. Most evidence comes from the United States, where findings have been heterogeneous and appear to vary by socioeconomic context, gender distribution, and occupational characteristics [20]. Teachers who develop chronic health issues not only increase their absenteeism [21,22], but also tend to experience a decline in their teaching performance. This decline can negatively impact both students and fellow teachers and complicate the planning efforts of school leaders. Ultimately, these factors generate unfavorable outcomes for the school and hinder student learning [23,24].

In Latin America, education faces persistent structural challenges, such as unequal access to quality services, underfunding, regional disparities, and high levels

of informality in the workforce [18,25]. In Colombia, this scenario is particularly relevant, as teachers constitute a large, diverse, and socially influential workforce that spans from preschool to university education. The educational sector is characterized by high demands, limited resources, and frequent sociopolitical pressures, all of which may exacerbate occupational risks and health inequalities among teachers [18]. Yet, despite their crucial role, evidence on teachers' health outcomes and occupational risks remains scarce. Existing studies have reported the prevalence of mental health conditions (such as stress and burnout), as well as voice, musculoskeletal, and vascular disorders; however, most of these investigations have focused on specific subgroups or local settings, leaving significant knowledge gaps at the national level [16,26,27].

In Colombia, the labor risk classification system assigns workers to one of five levels of occupational risk, ranging from level 1 (lowest risk) to level 5 (highest risk), depending on the nature of their job and associated hazards [28]. Teachers are generally classified under level 1, alongside other professions considered to have minimal physical risk (e.g., accountants, office clerks, secretaries [28]. Beyond risk classification, healthcare coverage also depends on the sector in which teachers work. Public school teachers are affiliated with a special insurance regime for public employees, which operates independently from the general system and has its own financing and management structure. In contrast, private school teachers are enrolled in the contributory regime, which covers nearly half of the national population (around 47%) and requires payroll-based contributions from both employees and employers, positioning it as one of the main financial pillars of the Colombian health system [29,30].

In this context, addressing teachers' health at a national scale is essential. This study used national administrative databases to describe and compare mortality and comorbidities between teachers and non-teacher workers classified in occupational risk level 1 in Colombia, and to examine differences in health outcomes across educational levels within the teaching profession. By focusing on private sector teachers, who are affiliated with the contributory scheme, and comparing them with other low-risk occupations, the study provides robust evidence to determine whether teaching is associated with distinctive health patterns that merit attention from both the education and health sectors. The findings contribute to filling a critical gap in occupational health research, offering novel insights into teachers' wellbeing in a Latin American context, and generating evidence to guide public health and occupational policies in the education sector.

## Methods

### Ethical considerations

This study was granted institutional review board (IRB) ethical approval by the Institutional Ethics Committee of Fundación Cardioinfantil – Instituto de Cardiología (Approval Number: CEIC-298–2025). Written consent was waived by the IRB as data sources were administrative databases fully anonymised. Following ethics approval, initial access to the databases was on 15/08/2025 with the purpose of identifying the population and creating a refined dataset for subsequent analysis. This refined dataset forms the basis of the analyses presented in this paper.

### Study design and population

A retrospective cohort study based on administrative databases was built with adults affiliated to the contributory health insurance scheme in 2017, whose occupational risk classification was coded as level 1 (lowest occupational risk). Individuals were identified using the Planilla Integrada de Liquidación de Aportes (PILA) database and subsequently classified as belonging to the teaching or non-teaching professions according to their reported occupational activity codes. Among those identified as teachers, we further categorized individuals by the educational level at which they worked, based on the specific occupation codes. Categories included preschool, primary, secondary, technical/technological education, university, and other education sectors. A detailed list of teaching profession codes used and their classification by educational level is provided in S1 File.

## Data and materials

This study utilized four primary administrative databases, anonymised and linked, using unique identifiers by the Ministry of Health (MoH). These databases were provided to the Clinical Research Institute at the Faculty of Medicine of the Universidad Nacional de Colombia by the Office of Information Technologies of MoH for research purposes (S2 File), and have been previously used in national studies because of their highly standardized [31–38]. Prior to their transfer, the Ministry anonymised the data by generating a unique identifier that allows for linkage across databases and enables longitudinal follow-up of individuals. The first source of information was the Wages Database (PILA), managed by the MoH, which contains detailed records on payroll contributions to health, pensions, and social security and precise information on formal employee wages monthly associated with their occupations and corresponding occupational risk classification associated with each activity [39]. The second source of information used was the Base for the Study of Capitation Unit Sufficiency (UPC in Spanish) database, which is the primary source of information used by MoH for the annual estimation of risk-adjusted capitated payment calculations and is populated by the largest insurers within the contributory scheme, covering approximately 19 million individuals (88% of this scheme's population nationwide). The UPC database includes detailed information on each utilized service, such as the Unified Health Procedure Code (CUPS in Spanish), an ICD-10 code associated with the service, the service date, the cost paid by each insurer to each provider, the patient's sex, the insurer to which the patient is affiliated, the city where the service was provided, the provider's registration code, and an anonymised individual identifier for each affiliate [40]. The third source of information was the Unique Enrollees Database (BDUA in Spanish) is the government registry for tracking demographic and enrolment information of all health system beneficiaries across contributory, subsidised, and other schemes [41]. It provides data on enrolment status, age, sex, insurer, and affiliation scheme. Finally, we used the Mortality Registry Module database (RUAF in Spanish), which contains information from death certificates for all Colombians, including sociodemographic characteristics and death information (i.e., date, cause and characteristics) [42]. International assessments have confirmed the reliability of RUAF data, with 91% of deaths registered through death certificates as of 2016 [43].

## Exposure, outcome, and control variables

To measure overall one-year mortality, we used information from the RUAF database, corresponding to the study's primary outcome variable. The secondary outcome was the presence of other comorbidities as mental illness and hearing illness, which were identified using the UPC database based on the registry of ICD-10 codes related to the conditions. We also included sociodemographic and clinical variables measured one year previously to control for potential confounders. Sociodemographic variables included age, sex, insurer, and department of residence. Clinical variables included the Charlson Comorbidity Index (CCI), validated for use in Colombia by Oliveros et al. [44], derived from ICD-10 codes. The CCI considered the presence of the following conditions: acute myocardial infarction, congestive heart failure, peripheral vascular disease, stroke, dementia, chronic pulmonary disease, connective tissue disease, peptic ulcer disease, liver disease, diabetes, complications of diabetes, cancer, metastatic cancer, paraplegia, renal disease, severe liver disease, and HIV.

## Analysis

Baseline characteristics of the cohort were described using measures of central tendency and dispersion for continuous variables and relative and absolute frequencies for categorical variables, stratified by occupation (teaching vs non-teaching). Among teachers, we identified specific educational subgroups using occupational activity codes, categorizing them as preschool, primary school, secondary school, academic high school, technical/technological education, university-level, and other education sectors. Crude one-year mortality, mental health disorder and hearing disorder proportions were estimated by sex, age group, and occupational category.

 

To assess factors associated with one-year all-cause mortality, mental health disorders, and hearing disorders, we fitted multivariable logistic regression models to estimate odds ratios (ORs) and their 95% confidence intervals for each outcome. Independent variables included in the models were sex, age, region of residence, number of months employed during the year, average wages, and insurer. Association estimates were first calculated comparing teachers versus non-teachers and then disaggregated by each educational level subgroup within the teaching profession. Statistical tests were performed to validate the assumptions of the independence of errors, heteroscedasticity, and multicollinearity for the regression model. All estimators were presented with 95% confidence intervals. Analyses were conducted using Stata 18.5 MP (licensed to the Universidad Nacional de Colombia) [45]. This article followed the STROBE (Strengthening the Reporting of Observational Studies in Epidemiology) guidelines to ensure transparency and thoroughness in reporting the findings, S3 File [46].

## Results

Between January 1, 2017, and December 31, 2017, 4,256,719 adults were affiliated with the contributory health insurance scheme in 2017 and reported occupational risk classification coded as level 1, of which 353,985 had teaching codes. The flowchart in Fig 1 provides additional details on the selection process. The mean age was 36.74 (SD 11.91) years, and

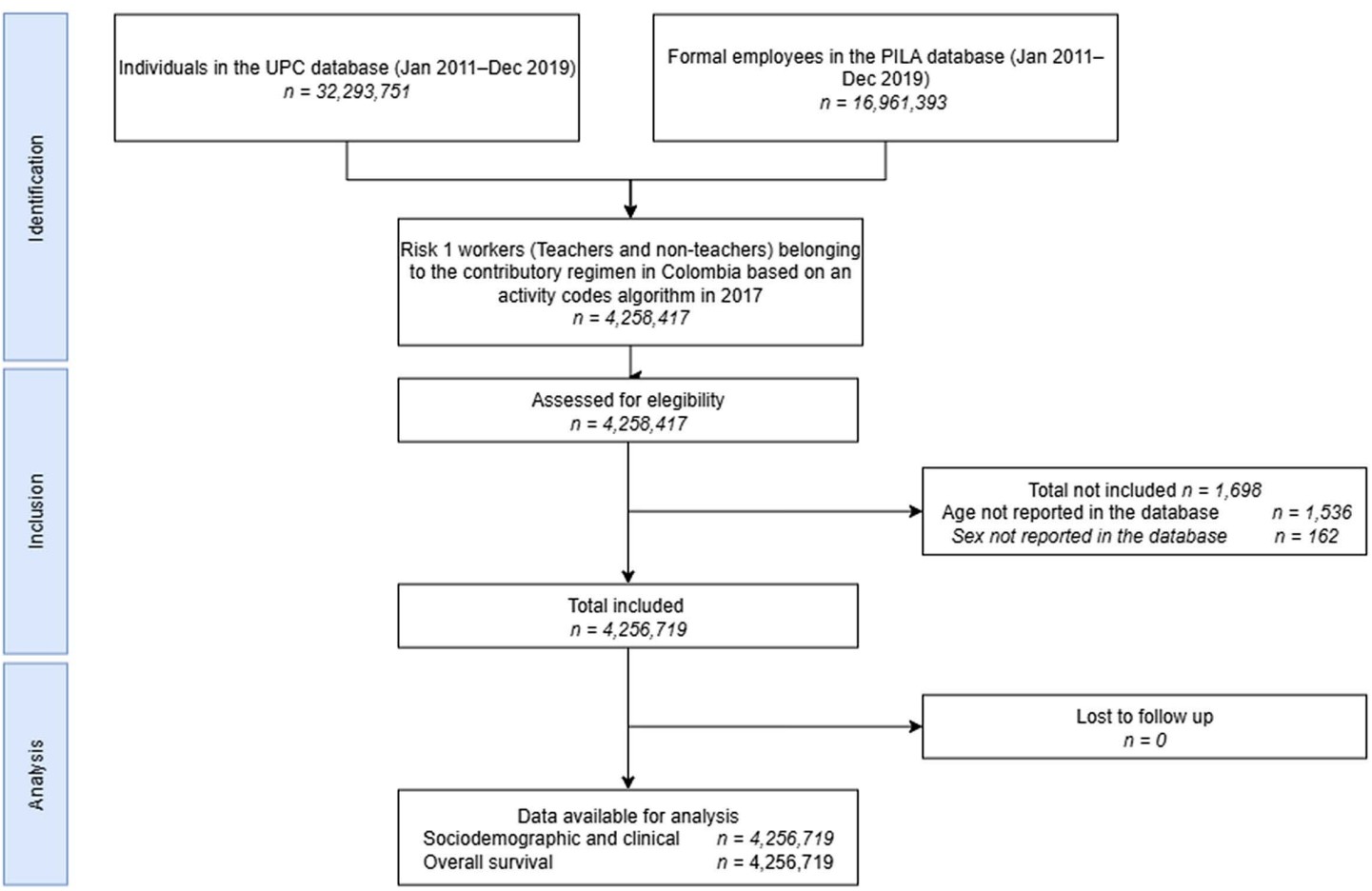

**Fig 1. Identification of population.** UPC: Base for the Study of Capitation Unit Sufficiency Database; PILA: Wages Database.

61.50% were female. 40.50% of the population identified residing in Bogotá D.C. 99.35% of the population had a CCI score of 0–3 points. Among the comorbidities evaluated, the most frequent reported were hypertension (9.21%), Diabetes Mellitus (2.14%) and Chronic Obstructive Pulmonary Disease (1.75%). Further details on the patients' sociodemographic and clinical characteristics are presented in Table 1.

Table 2 presents the proportion of one-year mortality, mental health disorders, and hearing disorders among individuals with teaching versus non-teaching occupations. Within the teaching group, outcomes were further stratified by the educational level in which individuals were employed. The overall mortality rate among teachers was 0.14 per 100 adults (95% CI, 0.13–0.15). Among teaching subgroups, the highest mortality proportions were observed in technical or technological education (0.21 per 100 adults, 95% CI 0.10–0.43) and in other education categories (0.15 per 100 adults, 95% CI 0.14–0.16). The proportion of individuals with a recorded mental health disorder was higher among those in teaching occupations overall, reaching 1·92 per 100 adults (95% CI 1·88–1·97). Within the teaching subgroups, the highest proportions were observed in primary education (1.78 per 100 adults, 95% CI 1.56–2.02) and other education sectors (1.99 per 100 adults, 95% CI 1.93–2.04). For hearing disorders, the highest proportions were reported among teachers in technical or technological education (0.56 per 100 adults, 95% CI 0.36–0.88) and in primary education (0.55 per 100 adults, 95% CI 0.44–0.70).

Table 3 presents the multivariate logistic models for 1-year mortality, mental illness and hearing illness, showing that teaching occupation was associated with a 15% more 1-year mortality compared to non-teaching occupation (OR: 1.15 95%CI: 1.03–1.27, p-value: 0.010) after adjusting for sex, age, CCI, geographic region, months worked, mean of wages and insurers. Other factors significantly associated with higher one-year mortality included male sex, older age, number of months employed, and lower average wages. No statistically significant associations were observed between teaching occupation and either mental health disorders or hearing disorders.

## Discussion

Our study found that teachers working in the private sector and classified within occupational risk level 1 in Colombia had a significantly higher risk of one-year mortality compared to other workers in the same category. This excess risk persisted after adjusting for sociodemographic and comorbidities factors. In contrast, no significant associations were observed between teaching occupation and either mental health disorders or hearing disorders.

When compared with previous literature, our results both align with and diverge from existing evidence. International studies have often highlighted the psychological burden of teaching, reporting elevated rates of burnout, stress, and depression among teachers [47–49]. Some Latin American studies have echoed these findings, noting a high prevalence of mental health issues in this profession [50–53]. However, most of these investigations have been limited to other countries outside Colombia or local settings, and few have provided robust comparisons against other occupational groups at the national level [16,26]. To our knowledge, this is the first study in Colombia to use large-scale administrative data to evaluate teachers' mortality and comorbidity profiles, offering a population-based perspective that complements and expands the existing body of research.

Several mechanisms may explain the observed increase in mortality among teachers. Occupational stress, long working hours, and limited resources in educational institutions may contribute to cumulative health risks, particularly cardiovascular and metabolic conditions [54]. The demographic profile of teachers, who in our study were older on average than non-teachers, may also partly account for the observed differences, although our models adjusted for age and comorbidities. It is also plausible that social and environmental exposures specific to the profession, such as high levels of psychosocial demand and job instability in certain sectors, may interact with broader determinants of health, producing a pattern of elevated mortality not mirrored in mental or hearing health outcomes [53,55,56].

A major strength of this study lies in the use of national administrative databases, which provided a large, population-based sample of teachers and non-teachers classified within the same occupational risk level. The

**Table 1. Sociodemographic characteristics of the population.**

| Variable | Total | Non-teachers | Teachers |
|---|---|---|---|
| | N = 4,256,719 | N = 3,902,734 | N = 353,985 |
| Age | 36.74 (11.91) | 36.43 (11.83) | 40.14 (12.25) |
| Categorized Age | | | |
| 18-40 yo | 2,643,325 (66.34%) | 2,457,967 (67.35%) | 185,358 (55.38%) |
| 40-60 yo | 1,202,341 (30.18%) | 1,071,476 (29.36%) | 130,865 (39.10%) |
| Older than 60 | 138,554 (3.48%) | 120,051 (3.29%) | 18,503 (5.53%) |
| Male | 1,681,049 (39.50%) | 1,571,218 (40.26%) | 109,831 (31.03%) |
| Date of start | | | |
| Region of residency[a] | | | |
| Atlántica | 393,172 (11.88%) | 360,806 (11.91%) | 32,366 (11.54%) |
| Bogotá DC | 1,340,464 (40.50%) | 1,232,924 (40.71%) | 107,540 (38.33%) |
| Central | 844,617 (25.52%) | 762,256 (25.17%) | 82,361 (29.36%) |
| Oriental | 277,999 (8.40%) | 253,868 (8.38%) | 24,131 (8.60%) |
| Pacífica | 437,301 (13.21%) | 404,276 (13.35%) | 33,025 (11.77%) |
| Orinoquía y amazonía | 15,878 (0.48%) | 14,772 (0.49%) | 1,106 (0.39%) |
| Comorbidities | | | |
| Myocardial infarction | 3,705 (0.14%) | 3,324 (0.14%) | 381 (0.17%) |
| Congestive heart failure | 3,778 (0.15%) | 3,348 (0.14%) | 430 (0.19%) |
| Periferical vascular disease | 1,744 (0.07%) | 1,574 (0.07%) | 170 (0.08%) |
| Stroke | 6,423 (0.25%) | 5,745 (0.24%) | 678 (0.30%) |
| Dementia | 822 (0.03%) | 715 (0.03%) | 107 (0.05%) |
| COPD | 45,342 (1.75%) | 40,948 (1.74%) | 4,394 (1.96%) |
| Connective tissue disease | 19,766 (0.76%) | 17,465 (0.74%) | 2,301 (1.02%) |
| Peptic Ulcer | 1,852 (0.07%) | 1,685 (0.07%) | 167 (0.07%) |
| Liver Disease | 1,075 (0.04%) | 965 (0.04%) | 110 (0.05%) |
| Hypertension | 237,972 (9.21%) | 211,236 (8.95%) | 26,736 (11.91%) |
| Diabetes Mellitus | 55,393 (2.14%) | 49,201 (2.08%) | 6,192 (2.76%) |
| Paraplegia | 543 (0.02%) | 495 (0.02%) | 48 (0.02%) |
| Kidney Disease | 14,097 (0.55%) | 12,401 (0.53%) | 1,696 (0.76%) |
| Cancer | 31,199 (1.21%) | 27,836 (1.18%) | 3,363 (1.50%) |
| Sever liver disease | 157 (0.01%) | 145 (0.01%) | 12 (0.01%) |
| HIV | 14,099 (0.55%) | 13,019 (0.55%) | 1,080 (0.48%) |
| Osteoarthritis | 129 (0.00%) | 115 (0.00%) | 14 (0.01%) |
| Mental Health Disorder | 71,047 (1.67%) | 64,234 (1.65%) | 6,813 (1.92%) |
| Hearing Disorder | 16,799 (0.39%) | 15,120 (0.39%) | 1,679 (0.47%) |
| CCI, median (IQR) | 0.00 (0.00) | 0.00 (0.00) | 0.00 (0.00) |
| CCI, mean (SD) | 0.13 (0.59) | 0.12 (0.59) | 0.14 (0.61) |
| CCI | | | |
| 0-3 | 2,567,647 (99.35%) | 2,344,614 (99.35%) | 223,033 (99.35%) |
| 4-6 | 15,541 (0.60%) | 14,188 (0.60%) | 1,353 (0.60%) |
| 7 or more | 1,343 (0.05%) | 1,236 (0.05%) | 107 (0.05%) |

Data are means (SD). [a]Region: Atlantica: Atlántico, Bolivar, Cesar, Cordoba, Magdalena, La Guajira, Sucre, and San Andres; Central: Antioquia, Caldas, Huila; Risaralda, Tolima, and Quindio; Oriental: Boyaca, Caquetá, Santander, Norte de Santander, Arauca, Meta, Casanare; and Pacífica: Valle del Cauca, Cauca, Choco, and Nariño; Orinoquía y Amazonia: Guainía, Amazonas, Guaviare, Vaupés and Vichada. Abbreviations: SD: standard deviation; p25: percentil 25; p75: percentil 75; CCI: Categorized Charlson Index.

**Table 2. Bivariate analysis of 1-year mortality, mental illness and hearing illness related to teaching profession.**

| Activity | 1-year mortality | | | Mental Illness | | | Hearing Illness | | |
|---|---|---|---|---|---|---|---|---|---|
| | % | [95% conf. interval] | | % | [95% conf. interval] | | % | [95% conf. interval] | |
| No-teachers | 0.11 | 0.10 | 0.11 | 1.65 | 1.63 | 1.66 | 0.39 | 0.38 | 0.39 |
| Teachers | 0.14* | 0.13 | 0.15 | 1.92* | 1.88 | 1.97 | 0.47* | 0.45 | 0.50 |
| Preschool | 0.10 | 0.06 | 0.15 | 1.63 | 1.47 | 1.81 | 0.36 | 0.29 | 0.45 |
| Primary education | 0.09 | 0.05 | 0.16 | 1.78 | 1.56 | 2.02 | 0.55* | 0.44 | 0.70 |
| Lower secondary education | 0.08 | 0.03 | 0.17 | 1.57 | 1.32 | 1.87 | 0.43 | 0.31 | 0.61 |
| Upper secondary education | 0.11 | 0.05 | 0.27 | 1.33 | 1.03 | 1.72 | 0.43 | 0.27 | 0.67 |
| Technical/technological education | 0.21 | 0.10 | 0.43 | 1.60 | 1.23 | 2.08 | 0.56 | 0.36 | 0.88 |
| Universities | 0.08 | 0.05 | 0.14 | 1.82 | 1.64 | 2.01 | 0.46 | 0.38 | 0.57 |
| Other education sectors | 0.15* | 0.14 | 0.16 | 1.99* | 1.93 | 2.04 | 0.48* | 0.46 | 0.51 |

**Table 3. Multivariate analysis of 1-year mortality, mental illness and hearing illness related to the teaching profession.**

| Variable | 1-year mortality | | | | Mental illness | | | | Hearing illness | | | |
|---|---|---|---|---|---|---|---|---|---|---|---|---|
| | Odds ratio | 95% CI lower | 95% CI upper | $P>|z|$ | Odds ratio | 95% CI lower | 95% CI upper | p | Odds ratio | 95% CI lower | 95% CI upper | $P>|z|$ |
| Non-Teachers | Ref | | | | Ref | | | | Ref | | | |
| Teachers | 1.15* | 1.03 | 1.28 | 0.010 | 0.98 | 0.96 | 1.01 | 0.146 | 0.97 | 0.92 | 1.02 | 0.257 |

Estimators were calculated adjusted by sex, age, region of residence, insurer, number of months employed, and lower average wages.

comprehensiveness and standardization of these datasets allowed us to evaluate outcomes across multiple educational subgroups and to adjust for relevant sociodemographic and clinical factors, ensuring the robustness of the findings. However, this study has several limitations. First, the use of administrative data constrained our ability to capture detailed clinical information, including behavioral risk factors (e.g., smoking, diet, physical activity), health behaviors, and psychosocial exposures such as workload, stress levels, or classroom conditions, all of which may be particularly relevant in the teaching profession. Second, the analysis focused on mortality, mental health disorders, and hearing disorders, thereby it left out other conditions that may disproportionately affect teachers, such as musculoskeletal disorders or voice problems. In addition, the one-year timeframe, although ensuring complete and consistent data availability, may not fully reflect longer-term health risks; studies with extended follow-up could offer a more comprehensive understanding. Furthermore, residual confounding may persist despite adjustment for multiple sociodemographic and clinical variables, as information on socioeconomic status, working conditions, and lifestyle factors was not available. The cross-sectional design further limits the ability to draw causal inferences. Finally, the study population was limited to teachers affiliated with the contributory regime, excluding those covered by the special regime in the public sector, which restricts the generalizability of the findings to the broader teaching workforce in Colombia.

Future research should seek to build on these findings by incorporating clinical and psychosocial data to better characterise the health risks of the teaching profession. Longitudinal studies would be particularly valuable to examine causal pathways linking occupational exposures to health outcomes. In addition, comparative analyses between public and private sector teachers, who are covered by different health insurance regimes, could shed light on the role of institutional arrangements in shaping health risks. Ultimately, generating a comprehensive evidence base on teachers' health is essential to inform occupational health policies and to strengthen the sustainability of the educational system in Colombia. Ultimately, there is a pressing need to deepen our understanding of how teachers' health affects student learning.

This relationship has been understudied and could be a focus for improving human capital accumulation and the overall well-being of society.

## Conclusion

This study provides novel evidence on the health of teachers in Colombia, showing that, despite being classified within the lowest occupational risk category, they experience a higher risk of mortality compared with other workers in the same group. No differences were observed in mental health or hearing disorders, suggesting that the most prominent disparities are reflected in mortality outcomes. These findings underscore the importance of strengthening efforts to improve health surveillance, prevention, and protection strategies focused on this population, ensuring that teachers' wellbeing is prioritized as part of broader occupational health and education policies in Colombia. Furthermore, our results can inform the design of programs or public policies aimed at reducing factors associated with health risks, which can contribute to improving not only teacher health but also their performance and reducing their absenteeism, ultimately leading to better student outcomes.

## Supporting information

**S1 File. Teaching activity codes.**
(DOCX)

**S2 File. Administrative database use approval.**
(PDF)

**S3 File. STROBE checklist.**
(DOCX)

## Author contributions

**Conceptualization:** Daniela Sánchez-Santiesteban, Hernando Bayona-Rodríguez, Giancarlo Buitrago.

**Data curation:** Daniela Sánchez-Santiesteban.

**Formal analysis:** Daniela Sánchez-Santiesteban.

**Methodology:** Daniela Sánchez-Santiesteban, Hernando Bayona-Rodríguez, Giancarlo Buitrago.

**Software:** Daniela Sánchez-Santiesteban, Giancarlo Buitrago.

**Writing – original draft:** Daniela Sánchez-Santiesteban, Hernando Bayona-Rodríguez, Giancarlo Buitrago.

**Writing – review & editing:** Daniela Sánchez-Santiesteban, Hernando Bayona-Rodríguez, Giancarlo Buitrago.

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
