## [Decision Letter · Decision Letter 0]

16 Oct 2025

Dear Dr. Sánchez-Santiesteban,

Thank you for submitting your manuscript to PLOS ONE. After careful consideration, we feel that it has merit but does not fully meet PLOS ONE’s publication criteria as it currently stands. Therefore, we invite you to submit a revised version of the manuscript that addresses the points raised during the review process.

We look forward to receiving your revised manuscript.

Kind regards,

Andres Acero

Academic Editor

PLOS ONE

Journal Requirements:

Additional Editor Comments:

Dear author(s):

Thank you on the comments of the reviewers, I consider that your contribution has potential to be published in PLOS ONE. However, there are some concerns from the reviewers that must be addressed before accepting the document. Please include the changes, and send it again for a next round with the reviewers.

Reviewer's Responses to Questions

**Comments to the Author**

1. Is the manuscript technically sound, and do the data support the conclusions?

Reviewer #1: Partly

Reviewer #2: Yes

2. Has the statistical analysis been performed appropriately and rigorously?

Reviewer #1: Yes

Reviewer #2: Yes

3. Have the authors made all data underlying the findings in their manuscript fully available?

Reviewer #1: Yes

Reviewer #2: Yes

4. Is the manuscript presented in an intelligible fashion and written in standard English?

Reviewer #1: Yes

Reviewer #2: Yes

Reviewer #1: Dear Authors,

I would like to commend you for selecting such a timely and important topic. The relationship between the teaching profession and health outcomes, including mortality and comorbidities, is a critical area of research that warrants further exploration. The study sheds valuable light on the health challenges faced by teachers, particularly in Colombia, and contributes to an ongoing discussion about occupational health risks.

However, there are several aspects you may want to consider to further enhance the depth and clarity of your research:

Research Questions: The research questions are not clearly defined in the paper, and as a result, the purpose of the statistical analysis is somewhat unclear. Defining the research questions upfront would not only guide the study's analysis but also help readers understand the scope and focus of the research. Clearly articulated research questions will enhance the transparency of the study's objectives.

Literature Review: The study would benefit from a more comprehensive literature review section. While you reference existing literature, it would be helpful to integrate a more detailed discussion of the findings from previous studies, particularly those that investigate similar health risks in other professions or geographical areas (in other continenets as well). Additionally, a review of best practices for addressing health risks in the teaching profession could provide more context and inform potential interventions.

Timeframe Justification: You use a one-year timeframe for your study, which is understandable but could be limiting. I would encourage to justify why you chose this particular timeframe. Is it possible to expand the study to a longer duration, say five years? A longer timeframe could add depth to the analysis, allowing for more robust conclusions regarding the long-term health risks of the teaching profession. This extension could also make the study's findings more solid and generalizable.

Ethics Section: I suggest moving the Ethics section to the end of the paper, as it does not align well with the Methods section. Ethical considerations are crucial, but they tend to follow the presentation of the study design and methodology. Placing it at the end would provide a more logical flow to the paper.

Limitations and Future Research: The study would benefit from a discussion on its limitations and potential areas for future research. Highlighting any limitations (e.g., the cross-sectional design, the generalizability of the findings) would provide a balanced perspective. Furthermore, suggesting avenues for future research could inspire further investigation into the health risks faced by teachers, perhaps in different geographical or cultural contexts. Without this, the study concludes somewhat abruptly, and a discussion of limitations would provide necessary nuance to the findings.

In conclusion, this study addresses an essential and underexplored issue, and the findings contribute to our understanding of the health implications of the teaching profession. With the suggested improvements, the paper would not only provide a clearer and more comprehensive analysis but also offer valuable insights for both policy and practice.

Best regards,

Reviewer

Reviewer #2: Reviewer’s Report

The manuscript requires revisions to improve clarity, organization, and scholarly rigor.

Specific Comments

1. Title Revision

The title should be rephrased for better clarity and precision. The current version can be refined as follows:

“Mortality and Comorbidities among Teaching Professionals: A Cross-Sectional Study in Colombia.”

This rephrasing maintains the original meaning while improving readability and academic tone.

2. Keywords

A few more relevant keywords should be included to enhance searchability and indexing. In addition to existing ones, consider adding terms such as occupational health, teachers, mortality rate, comorbidity patterns, and Colombian educators.

3. Introduction and Literature Review

The literature review needs to be extended. At present, it appears too brief and insufficiently contextualized. The introduction should include a more comprehensive discussion of prior studies related to occupational health risks, mortality, and comorbidities in the teaching profession—both globally and regionally. This will help establish the research gap and strengthen the rationale for the study.

4. Ethical Approval Statement

The ethical approval statement should not be included within the methodology section. It should be placed separately under a distinct subheading, such as “Ethical Considerations” or “Ethical Approval,” following standard research reporting conventions.

5. Data Transparency and Verification

Since the study utilizes four primary administrative databases anonymized and linked using 188 unique identifiers by the Ministry of Health (MoH), a hyperlink or citation should be provided to allow readers to verify the authenticity and source of these datasets. This step will enhance transparency and credibility.

The manuscript has potential for publication after major revisions. Addressing the above issues will significantly improve the clarity, scholarly depth, and transparency of the study.

**Do you want your identity to be public for this peer review?** For information about this choice, including consent withdrawal, please see our Privacy Policy

Reviewer #1: No

Reviewer #2: **Yes: ** Muhammad Ajmal

---

## [Author Response · Author response to Decision Letter 1]

24 Nov 2025

Manuscript ID: PONE-D-25-46538

Title: Mortality and Comorbidities among Teaching Professionals: A Cross-Sectional Study in Colombia

Authors: Daniela Sánchez-Santiesteban, Hernando Bayona-Rodríguez, Giancarlo Buitrago

Journal: PLoS ONE

Dear Editor and Reviewers,

We sincerely appreciate the time and expertise invested in reviewing our manuscript. We are grateful for the constructive and insightful comments, which have significantly strengthened the scientific rigor, clarity, and overall quality of the study. Following your recommendations, we have made substantial revisions to the manuscript and have addressed each point raised by the reviewers and the editorial team. Below, we provide a detailed, point-by-point response.

Reviewer #1

Comment 1. The research questions are not clearly defined in the paper, and as a result, the purpose of the statistical analysis is somewhat unclear. Defining the research questions upfront would not only guide the study's analysis but also help readers understand the scope and focus of the research. Clearly articulated research questions will enhance the transparency of the study's objectives.

- Response: Thank you for this helpful comment. We agree that clearly stating the research questions improves the transparency of our analysis. In the last paragraph of the Introduction, we now emphasize that the study aims to use national administrative databases to describe and compare mortality and comorbidities between teachers and non-teacher workers classified in occupational risk level 1 in Colombia, and to examine differences in health outcomes across educational levels within the teaching profession. We believe this clarification makes the objectives and scope of the statistical analysis more explicit for readers. The revised text reads (page 4, lines 154-158):

“This study used national administrative databases to describe and compare mortality and comorbidities between teachers and non-teacher workers classified in occupational risk level 1 in Colombia, and to examine differences in health outcomes across educational levels within the teaching profession.”

Comment 2: The study would benefit from a more comprehensive literature review section. While you reference existing literature, it would be helpful to integrate a more detailed discussion of the findings from previous studies, particularly those that investigate similar health risks in other professions or geographical areas (in other continenets as well). Additionally, a review of best practices for addressing health risks in the teaching profession could provide more context and inform potential interventions

- Response: We appreciate this thoughtful suggestion. In line with your recommendation, we have expanded the Introduction to provide a more comprehensive and contextualized literature review on occupational health risks, comorbidities, and mortality among teachers. Specifically, we now describe, page 3, line 108-151.

“Globally, research has shown that teachers may present distinct morbidity profiles relative to other workers, with a higher prevalence of mental health problems and hearing disorders, often linked to occupational stress and prolonged exposure to noisy environments [8–11]. Teachers are also exposed to physical and environmental hazards that contribute to other morbidities, such as ergonomic strain, poor air quality, and inadequate acoustic conditions further aggravate physical health outcomes [12]. In Latin America, growing evidence underscores a substantial burden of occupational morbidity among teachers. Studies from Brazil, Mexico, and Chile have consistently reported high rates of burnout, anxiety, and voice and hearing disorders associated with overcrowded classrooms, noise, and precarious working environments [13–15]. In Colombia, similar patterns have been observed, identifying a high prevalence of burnout syndrome, anxiety, and musculoskeletal complaints in the education sector [15–19]. Mortality patterns among teachers have been relatively understudied. Most evidence comes from the United States, where findings have been heterogeneous and appear to vary by socioeconomic context, gender distribution, and occupational characteristics [20]. Teachers who develop chronic health issues not only increase their absenteeism [21,22], but also tend to experience a decline in their teaching performance. This decline can negatively impact both students and fellow teachers and complicate the planning efforts of school leaders. Ultimately, these factors generate unfavorable outcomes for the school and hinder student learning [23,24].

In Latin America, education faces persistent structural challenges, such as unequal access to quality services, underfunding, regional disparities, and high levels of informality in the workforce [18,25]. In Colombia, this scenario is particularly relevant, as teachers constitute a large, diverse, and socially influential workforce that spans from preschool to university education. The educational sector is characterized by high demands, limited resources, and frequent sociopolitical pressures, all of which may exacerbate occupational risks and health inequalities among teachers [18]. Yet, despite their crucial role, evidence on teachers’ health outcomes and occupational risks remains scarce. Existing studies have reported the prevalence of mental health conditions (such as stress and burnout), as well as voice, musculoskeletal, and vascular disorders; however, most of these investigations have focused on specific subgroups or local settings, leaving significant knowledge gaps at the national level [16,26,27].

In Colombia, the labor risk classification system assigns workers to one of five levels of occupational risk, ranging from level 1 (lowest risk) to level 5 (highest risk), depending on the nature of their job and associated hazards [28]. Teachers are generally classified under level 1, alongside other professions considered to have minimal physical risk (e.g., accountants, office clerks, secretaries [28]. Beyond risk classification, healthcare coverage also depends on the sector in which teachers work. Public school teachers are affiliated with a special insurance regime for public employees, which operates independently from the general system and has its own financing and management structure. In contrast, private school teachers are enrolled in the contributory regime, which covers nearly half of the national population (around 47%) and requires payroll-based contributions from both employees and employers, positioning it as one of the main financial pillars of the Colombian health system [29,30].”

Comment 3: You use a one-year timeframe for your study, which is understandable but could be limiting. I would encourage to justify why you chose this particular timeframe. Is it possible to expand the study to a longer duration, say five years? A longer timeframe could add depth to the analysis, allowing for more robust conclusions regarding the long-term health risks of the teaching profession. This extension could also make the study's findings more solid and generalizable.

- Response: We appreciate this comment and agree that multi-year analyses could provide additional insights. We selected 2017 because it represents a large and nationally representative sample of teachers and non-teachers, and because all administrative datasets required for this study are fully available and complete for that year. This ensured consistent data coverage across sources and allowed us to conduct a comprehensive analysis aligned with the study objectives.

Comment 4: I suggest moving the Ethics section to the end of the paper, as it does not align well with the Methods section. Ethical considerations are crucial, but they tend to follow the presentation of the study design and methodology. Placing it at the end would provide a more logical flow to the paper.

- Response: Thank you for this suggestion. We have relocated the Ethics information to a dedicated subsection titled “Ethical Considerations,” which now appears after the Conclusions.

Comment 5: The study would benefit from a discussion on its limitations and potential areas for future research. Highlighting any limitations (e.g., the cross-sectional design, the generalizability of the findings) would provide a balanced perspective. Furthermore, suggesting avenues for future research could inspire further investigation into the health risks faced by teachers, perhaps in different geographical or cultural contexts. Without this, the study concludes somewhat abruptly, and a discussion of limitations would provide necessary nuance to the findings.

- Response: We agree that explicitly outlining the study’s limitations and future research directions is essential to provide a balanced interpretation of our findings. In the revised Discussion, we have expanded the limitations paragraph to highlight that the use of administrative data restricts the availability of detailed clinical and behavioral variables (e.g., smoking, diet, physical activity), that our outcomes were limited to mortality, mental health disorders, and hearing disorders, and that our analysis only includes teachers affiliated with the contributory regime, thus limiting generalizability to the entire teaching workforce in Colombia. We also emphasize that the cross-sectional design precludes causality. In addition, we have added a sentence on future research that underscores the need for longitudinal studies incorporating clinical and psychosocial data, as well as comparative analyses between public- and private-sector teachers covered by different insurance regimes. These additions provide a more nuanced discussion of our results and identify concrete avenues for further investigation. This limitations section appears on page 9, lines 341-366.

“However, this study has several limitations. First, the use of administrative data constrained our ability to capture detailed clinical information, including behavioral risk factors (e.g., smoking, diet, physical activity), health behaviors, and psychosocial exposures such as workload, stress levels, or classroom conditions, all of which may be particularly relevant in the teaching profession. Second, the analysis focused on mortality, mental health disorders, and hearing disorders, thereby it left out other conditions that may disproportionately affect teachers, such as musculoskeletal disorders or voice problems. In addition, the one-year timeframe, although ensuring complete and consistent data availability, may not fully reflect longer-term health risks; studies with extended follow-up could offer a more comprehensive understanding. Furthermore, residual confounding may persist despite adjustment for multiple sociodemographic and clinical variables, as information on socioeconomic status, working conditions, and lifestyle factors was not available. The cross-sectional design further limits the ability to draw causal inferences. Finally, the study population was limited to teachers affiliated with the contributory regime, excluding those covered by the special regime in the public sector, which restricts the generalizability of the findings to the broader teaching workforce in Colombia.

Future research should seek to build on these findings by incorporating clinical and psychosocial data to better characterise the health risks of the teaching profession. Longitudinal studies would be particularly valuable to examine causal pathways linking occupational exposures to health outcomes. In addition, comparative analyses between public and private sector teachers, who are covered by different health insurance regimes, could shed light on the role of institutional arrangements in shaping health risks. Ultimately, generating a comprehensive evidence base on teachers’ health is essential to inform occupational health policies and to strengthen the sustainability of the educational system in Colombia.”

Reviewer #2

Comment 1. Title revision: The title should be rephrased for better clarity and precision. The current version can be refined as follows: “Mortality and Comorbidities among Teaching Professionals: A Cross-Sectional Study in Colombia.” This rephrasing maintains the original meaning while improving readability and academic tone.

- Response: We appreciate this recommendation. We have revised the title to: “Mortality and Comorbidities among Teaching Professionals: A Cross-Sectional Study in Colombia,” as suggested.

Comment 2. Keywords: A few more relevant keywords should be included to enhance searchability and indexing. In addition to existing ones, consider adding terms such as occupational health, teachers, mortality rate, comorbidity patterns, and Colombian educators.

- Response: Thank you for this helpful suggestion. We have expanded the list of keywords to improve searchability and indexing, adding terms such as “occupational health,” “teachers,” “mortality rate,” “comorbidity patterns,” and “Colombian educators,” in line with your recommendation.

Comment 3. Expanded literature review: The literature review needs to be extended. At present, it appears too brief and insufficiently contextualized. The introduction should include a more comprehensive discussion of prior studies related to occupational health risks, mortality, and comorbidities in the teaching profession—both globally and regionally. This will help establish the research gap and strengthen the rationale for the study.

- Response: We agree with the reviewer that a more extensive and contextualized literature review is needed. Accordingly, we have strengthened the Introduction by expanding the discussion of comorbidities and adding evidence on mortality among teachers. As mentioned in a previous comment from reviewer 1 and indicated on page 3, line 108-151.

Comment 4. Ethics approval statement relocation

The ethical approval statement should not be included within the methodology section. It should be placed separately under a distinct subheading, such as “Ethical Considerations” or “Ethical Approval,” following standard research reporting conventions.

- Response: Thank you for this comment. We have revised the structure of the manuscript so that the ethical approval information appears in a separate subsection entitled “Ethical Approval,” placed after the Conclusions, rather than within the Methods section.

Comment 5. Data transparency

Since the study utilizes four primary administrative databases anonymized and linked using 188 unique identifiers by the Ministry of Health (MoH), a hyperlink or citation should be provided to allow readers to verify the authenticity and source of these datasets. This step will enhance transparency and credibility.

- Response: We appreciate this important comment regarding data transparency. The information sources used in this study (PILA, UPC, the Unique Enrollees Database, and RUAF) are administered by the Colombian Ministry of Health and Social Protection. These administrative databases are available to researchers upon request to the Office of Information and Communication Technologies at the Ministry (correo@minsalud.gov.co). However, under the terms of our data use agreement and national regulations, we are not allowed to redistribute or upload the individual-level datasets to public repositories. We have clarified this in the Data Availability Statement by specifying that the data are third-party administrative databases held by the Ministry of Health and can be accessed upon request to the authority, subject to the applicable legal and ethical requirements. We have also attached the official permission letter from the Ministry as supplementary material.

Editor Comments:

Comment 1: Thank you for uploading your study's underlying data set. Unfortunately, the repository you have noted in your Data Availability statement does not qualify as an acceptable data repository according to PLOS's standards. At this time, please upload the minimal data set necessary to replicate your study's findings to a stable, public repository (such as figshare or Dryad) and provide us with the relevant URLs, DOIs, or accession numbers that may be used to access these data. For a list of recommended repositories and additional inform

---

## [Decision Letter · Decision Letter 1]

18 Dec 2025

Mortality and Comorbidities among Teaching Professionals: A Cross-Sectional Study in Colombia

PONE-D-25-46538R1

Dear Dr. Sánchez-Santiesteban,

We’re pleased to inform you that your manuscript has been judged scientifically suitable for publication and will be formally accepted for publication once it meets all outstanding technical requirements.

Kind regards,

Andres Acero

Academic Editor

PLOS One

Additional Editor Comments (optional):

Dear Author(s),

We are pleased to inform you that your manuscript “Mortality and Comorbidities among Teaching Professionals: A Cross-Sectional Study in Colombia” has been accepted for publication in PLOS ONE.

Thank you for your careful revisions and for addressing the reviewers’ and editors’ comments. Your article will now proceed to production, and you will be contacted shortly with next steps.

Congratulations, and thank you for choosing PLOS ONE.

Reviewers' comments:

Reviewer's Responses to Questions

**Comments to the Author**

Reviewer #1: All comments have been addressed

Reviewer #2: All comments have been addressed

2. Is the manuscript technically sound, and do the data support the conclusions?

Reviewer #1: Yes

Reviewer #2: Yes

3. Has the statistical analysis been performed appropriately and rigorously?

Reviewer #1: Yes

Reviewer #2: Yes

4. Have the authors made all data underlying the findings in their manuscript fully available?

Reviewer #1: Yes

Reviewer #2: Yes

5. Is the manuscript presented in an intelligible fashion and written in standard English?

Reviewer #1: Yes

Reviewer #2: Yes

Reviewer #1: Dear Authors,

Thank you for thoroughly addressing the comments and questions raised during the first round of review. Your careful revisions are appreciated.

Best regards,

Reviewer

Reviewer #2: It is good to address changes suggested.It is good to address changes suggested.It is good to address changes suggested.It is good to address changes suggested.It is good to address changes suggested.

**Do you want your identity to be public for this peer review?** For information about this choice, including consent withdrawal, please see our Privacy Policy

Reviewer #1: No

Reviewer #2: **Yes: ** ajmal

---

## [Editor Report · Acceptance letter]

PONE-D-25-46538R1

PLOS One

Dear Dr. Sánchez-Santiesteban,

I'm pleased to inform you that your manuscript has been deemed suitable for publication in PLOS One. Congratulations! Your manuscript is now being handed over to our production team.

Kind regards,

on behalf of

Dr. Andres Acero

Academic Editor

PLOS One